# The Synergy between Technological Development and Logistic Cooperation of Road Transport Companies

**Kristina Vaičiūtė [1,*], Aušra Katinienė [2] and Gintautas Bureika [1]**

[1] Faculty of Transport Engineering, Vilnius Gediminas Technical University, Plytinės Str. 27, LT-10105 Vilnius, Lithuania
[2] Faculty of Fundamental Sciences, Vilnius Gediminas Technical University, Saulėtekio al. 11, LT-10223 Vilnius, Lithuania
* Correspondence: kristina.vaiciute@vilniustech.lt

**Abstract:** In today's world, with the acceleration of technological processes and the emergence of the wave of the COVID-19 pandemic, all production and service sectors, especially transport companies, are experiencing new challenges of technological development. The technological development of a transport company and the efficiency of road transport logistics depends on all the resources used and the latest implemented technologies. The authors analyse and evaluate the synergy of technological development and logistic cooperation. A synergy connection model is created to assess the interaction between logistic cooperation and technological development, and this synergy impacts the transport company's activity. Three groups of factors are arranged for the defining of the level of synergy of logistics cooperation and technological development of road transport companies: the first, the influence of technological development on the effective use of IT; the second, the influence of technological development on the quality of innovative equipment; and third, the impact of technological development on the level of staff competence. The Alternative Hierarchy Processing (AHP) method was chosen to assess these factors' importance as AHP enables quantifying the evaluation, i.e., gives numerical values for quality criteria. The synergy of transport company technological development and logistics cooperation is ensured mostly by the compatibility of transport system technologies and the technological literacy of logistic employees. It has been established that this factor is decisive for the formation of road transport companies' technological development and logistic cooperation processes. Finally, the conclusions and recommendations are given.

**Keywords:** road transport; technological development; logistic cooperation; transport company; sustainability; synergy; multi-criteria evaluation; AHP

## 1. Introduction

The growing needs of clients of transport companies and the rapidly expanding cooperation between transport companies oblige them to create and implement technological innovations instantly. Technological development of transport companies is one of the possible sources of innovation for the company's activities. Currently, with the globalising world, global economic competition, and rapidly developing conditions of informational cooperation in the transport sector, business companies that seek to maintain their competitiveness must keep up with accelerating market dynamics and improving technological development. It is therefore necessary to reconcile the technological means used in logistical cooperation for the transmission and sharing of information and to ensure the improvement of the performance of a road transport company.

The problem of technological development during the COVID-19 pandemic caused many discussions on ensuring cooperation between logistics companies, training processes in education, etc. It was significant for companies to have information systems to ensure

activity in a reduced business environment. In a short time, it was necessary to train in distantly the company's employees to manage these systems and use them productively during cooperation. Indian scientists Gupta et al. provided a Fuzzy Hybrid Multicriteria Decision-making (framework to identify society's vulnerability to COVID-19 transmission) [1]. Furthermore, to ensure comprehensive evaluation, researchers developed a three-phase framework consisting of the Fuzzy Delphi Method (FDM), Fuzzy Analytic Hierarchy Process (FAHP), and Similarity to Ideal Solution (FTOPSIS) methodologies. This framework enabled them to consider objective and subjective factors in the decision-making process simultaneously.

In another study, Indian scientists Rathore and Gupta proposed a wide-ranging safety risk set of psychological, ergonomic, and organisational, working conditions, as well as the associated tools, techniques, and safety risk categories in a fuzzy-based decision-making framework (DMF) [2]. This DMF enabled the identification of critical safety risk factors and prompted critical hospitals to improve their working conditions. The developed DMF creates a useful tool for helping the healthcare system solve problems. Nevertheless, this study [2] has some limitations, i.e., consideration of only ergonomic, psychological, work conditions, organisational and tool, technique, and technology factors.

France researchers Touratier and Jaussau considered 21 pilot transport company indicators; the 12 most relevant were presented during a brainstorming session with signatory enterprises [3]. Finally, the authors emphasised the need for more efficient inter-organisational collaboration between shippers and carriers. Thus, several indicators required a more transparent transmission of the data, particularly through truck telematics. Although carriers can access a large amount of information, a certain amount of disappointment is noticeable on the shippers' side as they face difficulties accessing this type of information.

Transport business owners are trying to enhance companies' value by developing growth strategies [4,5]. The CEO of the largest European asset-based road transport company, "Girteka Logistic", Mr. Liachovič, and co-authors, proposed a novel approach to solve the business valuation problem by considering both financial and non-financial drivers. A tremendous amount of digitalisation was required; IT implementation changes business models worldwide. Several Multicriteria Decision-Making (MCDM) methods selected the world-leading road freight transport companies for a case study. The Alternative Hierarchy Processing AHP method was used for driver weighting, and TOPSIS, COPRAS, SAW, PROMETHEE, and EDAS for the best alternative selection. The key drivers were identified: business model, the competence of employees and management system, and financial performance.

By including the logistics paradigm and IT, transport companies aim to enhance their responsiveness and flexibility [6]. The considered study [6] focuses on the organisational variables, IT capabilities, technological structures, and possible antecedents, and their impact on implementing logistic connections (LC). Researchers examined how LC implementation affects IT applications. The results showed that LC implementation has a mediating effect on IT use. Moreover, to the contrary, this study also revealed that IT capabilities have the most influence on LC implementation.

Companies are adopting innovative methods for responsiveness and efficiency in the logistics sector. One of these innovation options is drone delivery, which has the potential to change the traditional last-mile delivery process using trucks. Indian researchers Sah et al. identified and prioritised the barriers to drone logistics implementation based on their criticality by using FDM and AHP. Performed study [7] exposed that regulations and threats to privacy and security are the most critical barriers to implementing drones in logistics. Other important barriers are public perception, environmental issues, technical and economic aspects, and decreasing order of their criticality. In addition, other methods, such as the fuzzy technique for order preference by similarity to an ideal solution, fuzzy data envelopment analysis, and fuzzy analytic network process, can be used to compare the results.

Akbari [8] and Surdu et al. [9] analysed the characteristics of the business processes of road transport companies (hereinafter referred to as RTCs) and the factors influencing their cooperation. Sanchis-Pedregosa et al. [10], when analysing logistics processes and technological development factors, and their impact on RTC cooperation, quite superficially examined the impact of technological development on logistics cooperation. The RTC transportation service involves the transmission of information and the coordination of information in the supply chain [11] between companies that cooperate in logistics. Akbari [8], when analysing cooperation in the supply chain at the global level, emphasises the importance of effective management of logistics and transport information technology (IT) development. The impact of the service of land transport companies on cooperation and logistics activities comprises the process of managing the acquisition, movement and storage of goods or materials, parts and supplies [12]. The aim is to maximise the company's profitability by efficiently fulfilling orders and using innovative technologies to meet customer needs [8]. In order to provide high-quality transportation services and cooperation between transport or logistics companies, RTC staff must share information to use and properly select the latest technologies which enable technological development. RTC activities often lack synergies with other transport companies engaged in various modes of transport, i.e., effective communication channels are not formed in organising their transportation [13]. The importance of collaboration and the application of new effective technologies in business were examined by Arifin and Frmanzah [14], Hagiu and Altman [15] and Melville et al. [16]. The results of these studies confirmed that the application of new technologies in cooperation with other transport and/or logistics companies reduces operating costs and harmful effects on the environment and increases productivity. In 2018, the United Nations Intergovernmental Panel on Climate Change reported that road transport played one of the most important roles in the supply chain. According to the report [17], road transport emitted 72 % of the greenhouse gases (GHG) emissions of all transport. Therefore, the motivation of the research is to maximise the company's profitability by efficiently executing orders.

Furthermore, using innovative technologies to meet customer needs and at the same time contribute to $CO_2$ reduction is not limited to renewing the vehicle fleet used in the transport company. Implementing other means is also a significant factor, such as IT technologies, equipment quality and personnel competence level can be addressed, and thus perhaps there is a reduction in $CO_2$. The technological development of transport companies through cooperation between companies in the supply chain reduces GHG emissions and energy consumption if the technology is applied in different areas. This reduces fossil fuel consumption and ensures energy economy, which impacts $CO_2$ reduction [18]. However, there is a lack of more in-depth research to justify the importance of synergies between RTC technological development and collaboration and/or to identify processes.

## 2. Literature Review

### 2.1. Technological Development of Road Transport Companies

Technological development in logistics is a tool which helps to formulate, articulate and codify knowledge, i.e., to systematise laws and other normative acts [19]. The importance of business collaboration in technology development is highlighted in many studies [20–23]. Stone et al. state that cooperation and the quality of service are influenced by using and developing technologies [24]. RTC initiatives have shown the need for cooperation with each other to reduce pollution in cities. Nevertheless, this problem faces various organisational and technological challenges for cooperation [25–27]. According to Heitmann et al. [28], technology means the equipment and processes it performs to create and control the human world. New technology is usually installed in new objects, which means that the latter can be used in modern ways [29]. Technological development is needed for information exchange in RTC cooperation between logistics service providers, i.e., information must be provided quickly and efficiently. Transmission of infor-

mation by technological means, its efficient exchange between the RTCs applying different modes of transport and good technological information management is indispensable [30].

Researchers Glinskienė [31], Jones [32], and Jakubavičius et al. [33] describe the technological development in transport companies as constant work based on knowledge and practical experience that are designated to develop new products and equipment, to implement new processes, systems and services, and to substantially improve what has already been developed or implemented. Rakauskienė and Tamošiūnienė [34] connect the technological development of transport companies with the availability and use of information technologies and telecommunications in cooperation, as well as the availability and implementation of the up-to-date technologies. Adamauskas [35] states that technological development in logistics equals a renewal and change of technologies. The impact of transport companies' technological development on logistics cooperation is a complex process associated with a high risk to organisational systems, processes, products or even the industry itself. Håkansson and Waluszewski [36] define technological development as a continuous, albeit non-linear, process that allows for new directions in collaboration between RTCs in logistics. Crawford et al. [37] relate the technological development of logistics in collaboration with the IT infrastructure related to the technology itself, i.e., tools and resources that help collect, process, store, disseminate, use information and investments in exhibitions in IT results. Pérez-López and Alegre [38] present technological development as IT infrastructure development in transport companies, which consists of newly-acquired elements such as hardware, software and support staff. Technological development is an ongoing renewal process where RTC can take up new directions and services [36]. RTC, when collaborating with other logistics companies, especially those where firms seek joint development, shares knowledge and develop new competencies, and uses collaborative technologies. Collaborating companies must ensure that their partners have the appropriate technological capabilities or that the supplier can carry out technological developments that can help companies have preferential access to suppliers' resources [39]. Technological development is a process that depends on collaboration and interaction with different partners.

New promising areas are identified in analysing the aspects of technological development in RTCs, and the range of services provided to manage them is expanded [40]. Mammela et al. [41] argue that technological developments impact RTC profitability, economic growth, and resource efficiency. Innovation and technology affect employment through labour market flexibility, market competitiveness, types of innovation, the innovation system and international trade [42]. Sammalisto et al. [43], and Cruz and Sarmento [44] emphasise the impact of transport technology development management in logistics on workforce change, business competitiveness, survival, and local community development. Technological development changes communication and consumption habits in logistics, as there is a need to use improved operational methods in the activities, to use innovative technologies, but at the same time to be always receptive to further innovations. From the above, one can observe that new channels for the movement of goods are emerging in logistics, the innovation of employees is encouraged, and the attractiveness of certain companies to customers is growing [45]. The application of new technologies in RTC covers various areas. The technologies are like an axis on which other solutions that determine the coherence of the transport company's activities revolve [46]. The application of innovations and improved technologies in RTC activities change the economic situation itself. Acquisition of new knowledge and training of employees [47] should also include the ability to cooperate with parties to apply innovations and improve technologies in the process of knowledge acquisition. According to Kavadias et al. [48], technology development is usually associated with changes in the business model. The oncoming business model will reflect the connections between technological development and market needs. These interaction connections are presented in Figure 1 [43,44].

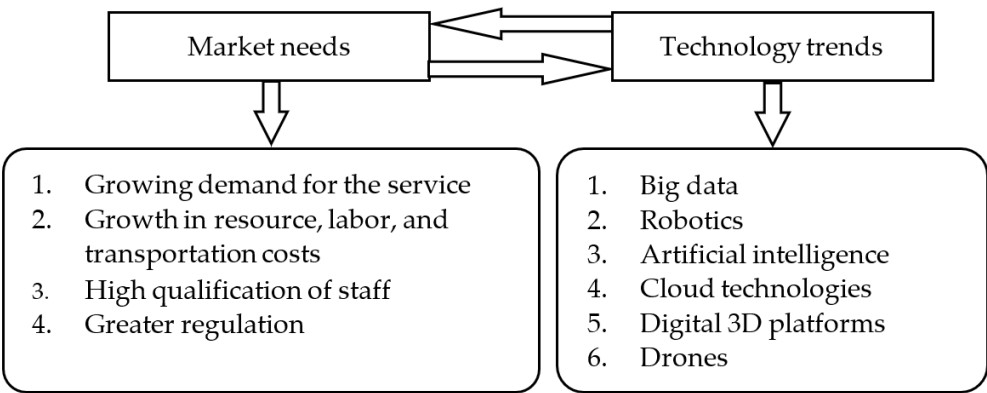

**Figure 1.** Connections between Logistic Cooperation and Technological Development [43,44].

In summary, it is assumed that the technological development of RTC is an innovative technological means for implementing new logistic/transportation processes and improving the existing ones related to the accessibility and availability of the use of information technologies and telecommunications.

RTC activities must be adequately adapted to technological developments and adjust the existing technology, as electronic means of communication in logistics operate so quickly that they turn information into products and services in seconds [16].

### 2.2. Logistic Cooperation of Transport Companies

According to David et al. [49], new technologies in logistics enable companies to "automate the routine tasks performed by employees and thus create a demand for a skilled workforce and foster employee innovation". The emerging interconnections between technology and information systems in RTC are becoming more important in achieving sustainability in logistics than each individual technology, as such interconnections have a greater impact on sustainability indicators [50]. A positive relationship between technological development changes in transport companies and employment in logistics is possible [51,52]. This is because new equipment, vehicles, machinery and production methods require a higher-skilled workforce [53], and technological progress offers opportunities to improve the quality of the service provided and, at the same time, to generate more revenue. Execution of technological development processes in logistics has these impacts on RTC activities:

1. For companies to access certain information through facilities, i.e., ecological sustainability and novelty of equipment in the sector and in transport activities [54];
2. For equipment reliability, i.e., non-perishability, longevity of operation, product quality assurance;
3. For the reliability of staff actions, i.e., the individual characteristics, qualifications and creativity of employees [55].

The structural scheme of RTC technological development's influence on the company activity digitalisation is presented in Figure 2.

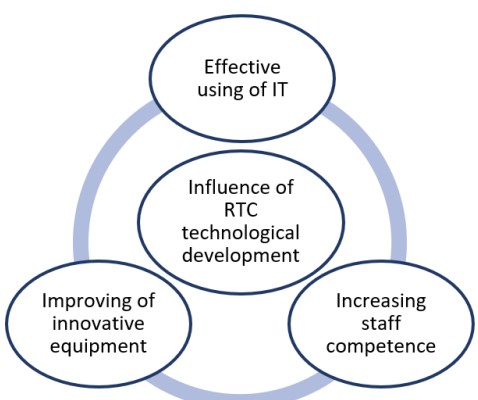

**Figure 2.** Influence of Technological Development on RTC activity digitalisation [52,54].

The factor of productivity of RTC service in logistics is highlighted by the need for technological development and improvement of human resources [56]. Productivity can be increased only by adapting to the constantly changing environmental conditions, constantly monitoring the market and applying technological developments in logistics properly [56,57]. Schwab and Sala-i-Martin [58] present three major subsystems based on the peculiarities of RTC technological development in RTC, which distinguish between:

(a)  Productivity promotion (higher education/courses, product/service market efficiency);
(b)  Labour market efficiency, development of financial markets, and market size;
(c)  Technological literacy of employees.

Transport market efficiency is achieved when all available asset-based resources and the newest technologies are used to provide the bulk of the service, ensuring the most intensive synergies of these factors. The synergies of RTC technological development and logistic cooperation are presented in Figure 3.

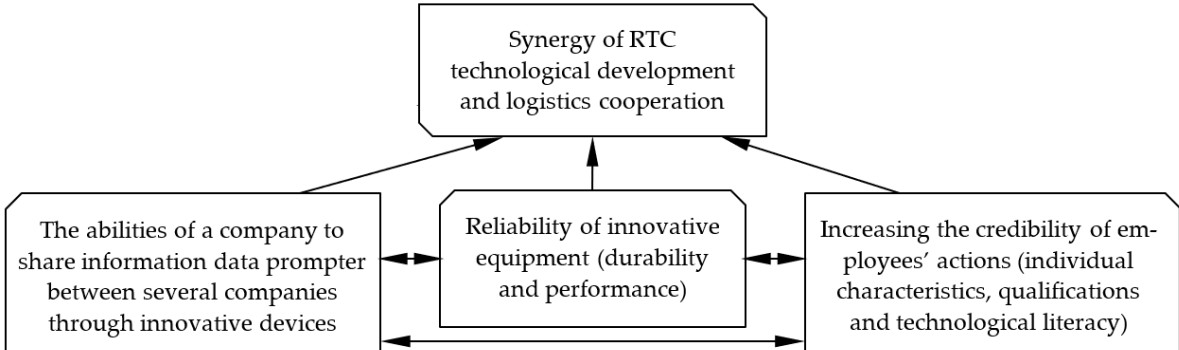

**Figure 3.** Synergy connections of RTC Technological Development and Logistic Cooperation [56,57].

RTC, by cooperating in logistics and exchanging information with the technologies operating in the companies, must ensure that the partners or customers have technological opportunities and can develop technologies so that the companies of customers and suppliers could gain access to information resources [39]. Wangai et al. [59] argue that future technologies will improve the efficiency and safety of transport logistics and reduce the environmental impact of future transport systems. The study of these technological effects requires the development of general approaches at system level and specific methods at subsystem, or vehicle, level. Efficient freight transport and cooperation between transport companies in logistics require closer coordination of transport infrastructure, including customs, airports, ports, railways and roads. Advanced technologies are constantly being introduced for freight transport companies' logistics cooperation and transport management [60].

In summary, it is concluded that the technological development of RTC is part of the transport logistics process, which depends on cooperation and interaction with different partners. Based on the literature review, the links of logistics cooperation with the development of RTC technologies can be grouped (to Figure 3): (1) impact of technological development on efficient use of IT; (2) impact of technological development on the quality of innovative equipment; (3) impact of technological development on the level of staff competence.

This research work aims to justify the synergies of technological development and logistic cooperation of RTC. The object of this research was the connections between RTC development and market needs in freight transportation processes. The authors have developed a methodology for evaluating the interfaces between the RTC technological development cooperation and the capacity of freight logistics lines. The evaluation criteria of RTC technological development and transportation logistics processes were identified during this research. The authors evaluated the impact of RTC technological development on the efficiency of transport logistics using a multi-criteria evaluation method, especially Alternative Hierarchy Processing (AHP). The consistency of expert opinions was determined by calculating the Kendall rank correlation coefficient. Analysis and synthesis methods were used to consider theoretical aspects and to process the outcomes.

## 3. Methodology

Researchers Stojčič et al. considered the problem concerning the application of MCDM methods in sustainable engineering [61]. They analysed 108 papers from 2008-2018, of which 23 papers were from Transport and Logistics. The results of this study show that sustainable engineering is an area that is quite suitable for the use of MCDM, first of all, approaching by applying the theory of uncertainty: fuzzy, grey rough and neurotrophic.

AHP is widely applied in various fields such as planning, decision-making, selecting the best alternative, resource allocations, resolving conflict, optimisation, etc. and numerical extensions of AHP [62]. According to the study's authors [62] primary group of AHP application fields are selection, evaluation, benefit-cost analysis, allocations, planning and development, priority and ranking, and decision-making.

Another benefit of AHP for evaluating technological development is an accurate approach to quantifying (giving a numerical value for quality norms) the weights of decision criteria [63–65]. Decision situations to which the AHP can be applied include:

1. Choice. The selection of one alternative (solution) from a given set of alternatives (solutions), usually where there are multiple decision criteria involved;
2. Ranking. Putting a set of alternatives in order from most to least significant;
3. Prioritisation. Determining the relative value of members of a set of alternatives instead of selecting a single one or just ranking them;
4. Resource allocation. Apportioning resources among a set of alternatives;
5. Benchmarking. Comparing the processes in one's organisation with those of other best-of-breed organisations;
6. Quality management. Dealing with the multi-criteria aspects of quality and quality improvement;
7. Conflict resolution. Settling disputes between parties with apparently incompatible goals or positions.

The authors selected the AHP method to assess the impact of RTC technological development on logistical cooperation and freight organisation [66]. The AHP method allows the conversion of qualitative evaluation criteria into quantitative criteria, i.e., enables the digitisation of the evaluation process. AHP provides a hierarchical approach to obtaining information about alternatives using pairwise comparisons of alternatives [67,68]. The AHP method is based on a pairwise comparison matrix. Experts compare everyone evaluated, assigning indicators $R_i$ and $R_j$ ($i, j = 1, \ldots, m$), ($m$—number of indicators). With the help of the method, qualitative evaluations of experts' indicators are rearranged into quantitative ones, i.e., indicator weights. The result of the comparison is square matrix $\boldsymbol{P} = (p_{ij})$

($i, j = 1, \ldots, m$). The elements $p_{ij}$ of the matrix, $\boldsymbol{P}$ are treated as ratios of the weight values of the indicators $R_i$ and $R_j$, when $p_{ij} = \frac{1}{p_{ji}}$.

One of the advantages of the AHP method over other multi-criteria assessment methods is its flexibility, convenience for decision-makers and the ability to check for compatibility [69]. Another positive feature is the possibility of evaluating the qualitative as quantitative indicators, with definite criteria significances. The advantage of the Analytical Hierarchy Process method is influenced by its ability to evaluate the decisions of experts and project stakeholders by selecting criteria according to hierarchical levels; this approach offers a better applicability to evaluating qualitative indicators. The Concordance coefficient can quantify the consistency of the opinion of several experts.

The Concordance coefficient indicates the level of coherence of the expert group if the number of experts is greater than two. Expert assessments obtained from the completed questionnaires are listed in the table. Based on the AHP methodology, the expert group $n$ quantifies objects $m$. Based on the Multi-Criteria Evaluation Method, the evaluations constitute a matrix of $n$ rows and $m$ columns and are provided in Table 1 [70]. The evaluation can be fulfilled in indicator units, unit fractions, percentages or in a decimal system. The ranking of expert indicators is suitable for calculating the Concordance coefficient. The ranking is a procedure in which the most important indicator is given a rank of $R$, equal to one, the second indicator is given a second rank, and the last indicator is given a rank $m$ ($m$ is the number of benchmarks).

**Table 1.** Matrix of evaluation indicators (compiled by the authors according to Sivilevičius [70]).

| Expert Code | | Indicator Marker, $j$ = 1, 2, ... , $m$ | | | | |
|---|---|---|---|---|---|---|
| | | $X_1$ | $X_2$ | $X_3$ | ... | $X_m$ |
| | $E_1$ | $R_{11}$ | $R_{12}$ | $R_{13}$ | ... | $R_{1m}$ |
| | $E_2$ | $R_{21}$ | $R_{22}$ | $R_{23}$ | ... | $R_{2m}$ |
| $i$ = 1, 2, ... , $n$ | $E_3$ | $R_{31}$ | $R_{32}$ | $R_{33}$ | ... | $R_{3m}$ |
| | ... | ... | ... | ... | ... | ... |
| | $E_n$ | $R_{n1}$ | $R_{n2}$ | $R_{n3}$ | ... | $R_{nm}$ |

The average of the sums of the ranks is calculated [71]:

$$\sum_{i=1}^{n} R_{ij} - \frac{1}{2}n(m+1);$$ (1)

where: $R_{ij}$—a rank of $R$, $m$—is the number of benchmarks, $n$—the number of experts.

$$W = \frac{12S}{n^2 m(m^2-1)} = \frac{12S}{n^2(m^3-m)}$$ (2)

where: $W$—the Concordance coefficient, $S$—the sum of the squares of the deviation from the arithmetic mean. Pearson criterion $\chi^2$ is calculated by the formula:

$$\chi^2 = n(m-1)W = \frac{12S}{nm(m+1)}$$ (3)

The minimum value of the Concordance coefficient $W_{\min}$ is determined by:

$$W_{\min} = \frac{\chi^2_{v,\alpha}}{n(m-1)}$$ (4)

$$S_{\max} = \frac{n^2 m(m^2-1)}{12};$$ (5)

$$\overline{R} = \frac{1}{2}n(m+1). \tag{6}$$

In light of the experts' evaluation indicators (6), the consistency of their opinions is determined by calculating the Concordance coefficient of the Kendall ranks. Suppose the variance S is a real sum of squares calculated according to Formula (1). In that case, the concordance coefficient by Formula (2), in the absence of related ranks, is defined by the ratio of the resulting $S$ to the corresponding maximum $S_{max}$ by Formula (5). The threshold value for the concordance coefficient is where expert assessments can be considered coordinated and the significance of the concordance coefficient can be determined using the Pearson criterion $\chi^2$ by Formula (3). The lowest value of the Concordance coefficient $W_{min}$ is calculated by Formula (4).

The competency coefficient of each expert is calculated using the method of calculating the expert competence coefficient [72]. In Formula (7), at the first iteration, all experts are given the same competence coefficient [73]. Giving the same weight to all experts shows whether the views of the experts are unanimous and competent. For this purpose, the competence coefficient of experts is calculated:

$$K_j^0 = \frac{1}{n}, \ j = 1, \ldots, n \tag{7}$$

where: $n$—number of experts, $K_j^0$—expert competence coefficient, 0—index of the first iteration equal to 1.

The sums of the initial rank values in the columns are then multiplied by the initial competency coefficient. Group estimates of alternatives (8) and a new matrix for calculating the competence coefficient were obtained. The competence coefficient [73] is calculated according to Formulas (8)–(10):

$$X_j^t = \sum_{i=1}^{m} K_i^{t-1} \cdot x_{ij}, \ j = 1, \ldots, n; \tag{8}$$

where: $X_j^t$—new matrix values; $\sum_{i=1}^{m} K_i^{t-1}$—group assessments; $x_{ij} = \ldots$; $i$—number of experts; $j$—the rank of the alternative.

$$\lambda^t = \sum_{j=1}^{n} \sum_{i=1}^{m} x_j^t \cdot x_{ij}; \tag{9}$$

where: $\lambda^t$—which is all matrices—$x_j^t$- the sum of the values.

$$K_i^t = \frac{1}{\lambda^t} \cdot \sum_{j=1}^{n} x_j^t \cdot x_{ij}, \quad \sum_{i=1}^{m} K_i^t = 1. \tag{10}$$

In the direct method weighting of criteria, $c_{ik}$ the sum of the weights of all the evaluations of each expert must be equal to 1.0 (or 100%). The method used to determine the weights of the criteria indirectly uses the chosen scoring system (5, 10, 20, etc.). Evaluations may be repeated. The weights $w$ of the criteria are calculated by the direct and indirect estimation method according to the Formula [71]:

$$w = \frac{\sum_{k=1}^{r} c_{ik}}{\sum_{i=1}^{m} \sum_{k=1}^{r} c_{ik}} \tag{11}$$

Expert evaluations are noted as $c_{ik}$ ($i = 1, \ldots, M; k = 1, \ldots, R$), where $m$—the number of applied criteria, $r$—the number of experts. The research process is based on several statements (hypotheses), which can be used, applying the AHP method, to determine the impact of technological development on the increase of RTC logistical cooperation.

Hypotheses T1, T2 and T3, should be confirmed or refuted. Hypotheses are formulated, having checked the consistency of expert opinions.

According to Yun [43], the interconnection of technology and information systems in transport companies has become more important to achieve the coherence of cooperation in logistics. This hypothesis was raised:

T1: The expectations for the use of RTC logistics collaboration technologies are higher than the technologies currently used.

According to Lovelace et al. [50], the feasibility of technological development processes in logistics could mean the possibilities of the companies to obtain information through interconnected facilities and the ability to manage the information received. This hypothesis was raised:

T2: The possibilities of RTC logistics cooperation require the compatibility of technology development with the technologies used between several companies to achieve a larger possible flow of information.

Vivarelli [57] argues that the ability of employees to adapt to the ever-changing collaboration and technological environment and applying the technology development in logistics would lead to productivity gains if knowledge of the technological elements is present. The third hypothesis raised was:

T3: Employee competencies in managing and controlling the newest technology elements are important in RTC collaboration in freight logistics.

In order to perform the evaluation of the RTC activity factors and to check, an algorithm consisting of six iterations was developed. In the first iteration, two steps are performed in parallel: model analysis and factor refinement. The second iteration is for preparation, i.e., selection of experts. It sets out the criteria and methods for selecting experts. The third iteration is for conducting the survey, i.e., for an interview with experts. The main concepts and peculiarities of completing the questionnaire were explained during this stage. In the next, the fourth iteration, the compatibility of the experts' opinions and the determination of their competencies are calculated. In the absence of a consensus, the third iteration is carried out, in the case of the fifth iteration, in which the results are processed, and decisions are taken. In the sixth iteration, the model is adjusted, i.e., according to the priorities of the expert factors, the most important factors are integrated into the proposed model. Carrying out a model (Figure 2) correction and including priority factors in it would increase the completeness of evaluating of synergy between technological development and logistical cooperation, i.e., strengthen employee communication and cooperation between various RTCs, open more opportunities for the use and quality improvement of innovative equipment, and streamline IT use processes. It would be necessary to eliminate or add untimely factors from the model, considering the analysis of the experts' results. The questionnaire uses questions with the options of answers provided, allowing choosing the answer. The minimum number of experts to form an expert group is three, and it is recommended to include at least five. In order to increase the reliability of the expert assessment, as many researchers argue, the optimal size of the expert group is from eight to ten experts; here, eight experts were included [74,75].

## 4. Results

During the evaluation of the impact of RTC technological development on logistical cooperation, the evaluation criteria were divided into three groups according to:

Group 1: Impact of technological development on efficient use of IT;
Group 2: Impact of technological development on the quality of innovative equipment;
Group 3: Impact of technological development on the level of staff competence.

In order to evaluate the impact of RTC technological development on logistical cooperation, an algorithm consisting of four stages was developed. The First Stage is designated for preparation, i.e., a questionnaire is drawn up. In the Second Stage, experts were selected according to their qualifications, education and practical experience gained by the experts. Experts were introduced to the course of the research. It was emphasised that the study

would be only about RTC logistics specialists. In the Third Stage, an expert survey was carried out, and the compatibility of data matrices and expert opinions was calculated, which was determined by calculating the Concordance coefficient of Kendall ranks. If the opinions are reconciled, the Fourth Stage should be carried out. If there is a disagreement, then the experts are interviewed again, and if the opinions are conciliated, the Fourth Stage data processing step is carried out. The ranking method is used to identify $m$ objects quantified by the expert group $n$.

In order to make the results of the survey representative, a target segment of the survey competence was identified: an expert with at least three years of experience, with at least a master's degree, an intermediate manager at an RTC, working in forwarding or logistics departments.

The study involved eight experts from different RTC divisions and branches, all with 3 to 10 years of RTC leadership experience and all with the education to the level of master's degrees.

Experts (hereafter E) had to assess the influence of the synergy criteria of RTC technology development on the quality of transport, applying AHP and ranking methods (in order of importance: one as the most important, nine as the least important). The technological development criteria of the Road Transport Company (hereinafter RTC) for logistical cooperation in freight transport services are:

K1: accessibility to data through innovative devices;
K2: reliability of equipment used;
K3: reliability (adequacy) and computer literacy of the employees' actions;
K4: promoting technological development;
K5: technological literacy of employees;
K6: incompatibility of technology between road transport and other modes of transport;
K7: compatibility of collaborative technological developments of several RTCs;
K8: ignorance of new technology management;
K9: lack of funds for technological development.

The AHP method prioritises using a pairwise comparison method with some relative importance, which is reflected in Table 2.

**Table 2.** Assessment Score Scale from 1 to 9 of the AHP method (compiled by the Authors according to Saaty [66]).

| Level of Importance | Definitions |
| --- | --- |
| 1 | Indicators are equally important |
| 3 | One indicator is slightly more important than the other |
| 5 | One indicator is more important than another |
| 7 | One indicator is much more important than another |
| 9 | One indicator is absolutely more important than another |
| 2, 4, 6, 8 | Intermediate values |

A pairwise comparison data of Expert 1 is presented in Table 3. Expert 1 was selected because the weighting criteria for this expert's assessment are the closest to the average criterion weights.

**Table 3.** Setting the Priorities of the 1st Expert Evaluation with the AHP method (compiled by the Authors).

| Criteria | K5 | K2 | K1 | K3 | K4 | K6 | K7 | K9 | K8 |
|----------|----|----|----|----|----|----|----|----|----|
| K5 | 1 | | | | | | | | |
| K2 | 2 | 1 | | | | | | | |
| K1 | 3 | 2 | 1 | | | | | | |
| K3 | 4 | 3 | 2 | 1 | | | | | |
| K4 | 5 | 4 | 3 | 2 | 1 | | | | |
| K6 | 6 | 5 | 4 | 3 | 2 | 1 | | | |
| K7 | 7 | 6 | 5 | 4 | 3 | 2 | 1 | | |
| K9 | 8 | 7 | 6 | 5 | 5 | 3 | 3 | 1 | |
| K8 | 9 | 8 | 7 | 6 | 5 | 4 | 3 | 3 | 1 |

A pairwise comparison is performed, which is necessary to find criteria for weights and significance of dependencies.

Table 4 provides an example of the first expert pairwise comparison matrix Equation (12). The square matrix comparison $P = (p_{ij})$ $(i, j = 1, \ldots, m)$. The elements $p_{ij}$ of the matrix $P$ are treated as ratios of the weight values of the indicators $R_i$ and $R_j$:

$$p_{ij} = {}^1/p_{ji} = p_{41} = {}^1/K_{14} = 1/3 = 0.33 \tag{12}$$

Elements are compared according to dominance or influence relationships with elements at a lower level. Connections of the relationships indicate the direction in which subordination connections operate.

**Table 4.** Pairwise comparison matrix of the first expert with AHP method (compiled by the Authors).

| Criteria | K1 | K2 | K3 | K4 | K5 | K6 | K7 | K8 | K9 | Geometric Mean | Weight of the 1st Expert Assessment |
|----------|----|----|----|----|----|----|----|----|----|----------------|-------------------------------------|
| K1 | 1 | 0.5 | 2 | 3 | 0.33 | 4 | 5 | 7 | 6 | 2.113 | 0.157 |
| K2 | 2 | 1 | 3 | 4 | 0.5 | 5 | 6 | 8 | 7 | 3.008 | 0.223 |
| K3 | 0.5 | 0.33 | 1 | 2 | 0.25 | 3 | 4 | 6 | 5 | 1.459 | 0.108 |
| K4 | 0.33 | 0.25 | 0.5 | 1 | 0.2 | 2 | 3 | 5 | 5 | 1.025 | 0.076 |
| K5 | 3 | 2 | 4 | 5 | 1 | 6 | 7 | 9 | 8 | 4.147 | 0.308 |
| K6 | 0.25 | 0.2 | 0.33 | 0.5 | 0.167 | 1 | 2 | 4 | 3 | 0.685 | 0.051 |
| K7 | 0.2 | 0.17 | 0.25 | 0.33 | 0.143 | 0.5 | 1 | 3 | 3 | 0.495 | 0.037 |
| K8 | 0.14 | 0.13 | 0.17 | 0.2 | 0.125 | 0.33 | 0.33 | 1 | 0.33 | 0.241 | 0.018 |
| K9 | 0.17 | 0.14 | 0.20 | 0.2 | 0.111 | 0.25 | 0.33 | 3 | 1 | 0.310 | 0.023 |
| Sum | 7.59 | 4.72 | 11.45 | 16.23 | 2.83 | 22.08 | 28.67 | 44.00 | 40.33 | 13.484 | 1.00 |

According to the first expert's internal significance of factor groups, a matrix of internal dependencies is formed, according to which the significance of factor groups obtained by the AHP method is recalculated (Table 5).

**Table 5.** Significance of Expert Assessment with AHP method (compiled by the authors).

| Criteria | E1 | E2 | E3 | E4 | E5 | E6 | E7 | E8 |
|---|---|---|---|---|---|---|---|---|
| K1 | 0.157 | 0.024 | 0.016 | 0.024 | 0.018 | 0.024 | 0.017 | 0.024 |
| K2 | 0.223 | 0.039 | 0.024 | 0.039 | 0.038 | 0.039 | 0.024 | 0.039 |
| K3 | 0.108 | 0.051 | 0.051 | 0.051 | 0.051 | 0.051 | 0.051 | 0.051 |
| K4 | 0.076 | 0.076 | 0.039 | 0.076 | 0.022 | 0.076 | 0.037 | 0.076 |
| K5 | 0.308 | 0.108 | 0.307 | 0.108 | 0.108 | 0.108 | 0.108 | 0.108 |
| K6 | 0.051 | 0.307 | 0.076 | 0.307 | 0.307 | 0.307 | 0.223 | 0.307 |
| K7 | 0.037 | 0.223 | 0.108 | 0.223 | 0.157 | 0.223 | 0.307 | 0.223 |
| K8 | 0.018 | 0.156 | 0.156 | 0.156 | 0.223 | 0.156 | 0.157 | 0.156 |
| K9 | 0.023 | 0.016 | 0.223 | 0.016 | 0.076 | 0.016 | 0.076 | 0.016 |
| Sum | 1 | 1 | 1 | 1 | 1 | 1 | 1 | 1 |

Consistency of opinions with more than two experts is checked by Concordance coefficients. When calculating the Concordance coefficient, expert evaluations are ranked. The distribution of the ranking of technological development criteria by importance (rank one is the most important, rank is the least important) is presented in Figure 4.

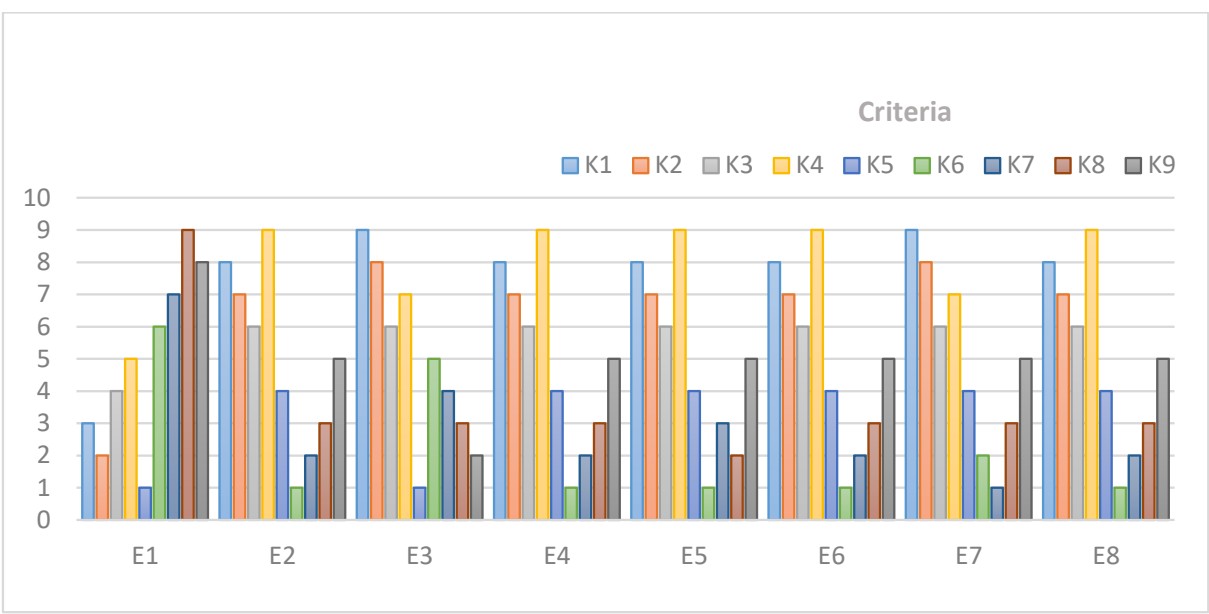

**Figure 4.** Distribution of the ranking of criteria of technological development (compiled by the authors).

The data of the analysis and calculation of the distribution of the rankings of eight experts' criteria are listed in Table 6.

**Table 6.** Ranking results of the importance of RTC technological development elements for logistics cooperation (compiled by the authors).

| The Mathematical Expression of the Criterion | Criterion Encryption Symbol ($m = 9$) | | | | | | | | |
|---|---|---|---|---|---|---|---|---|---|
| | **K1** | **K2** | **K3** | **K4** | **K5** | **K6** | **K7** | **K8** | **K9** |
| $\sum\limits_{i=1}^{n} R_{ij}$ | 61 | 53 | 46 | 64 | 26 | 18 | 23 | 29 | 40 |
| $\overline{R}_j = \frac{\sum_{i=1}^{n} R_{ij}}{n}$ | 7.625 | 6.625 | 5.75 | 8 | 3.25 | 2.25 | 2.875 | 3.625 | 5 |
| $\sum\limits_{i=1}^{n} R_{ij} - \frac{1}{2}n(m+1)$ | 21 | 13 | 6 | 24 | −14 | −22 | −17 | −11 | 0 |
| $\left[\sum\limits_{i=1}^{n} R_{ij} - \frac{1}{2}n(m+1)\right]^2$ | 441 | 169 | 36 | 576 | 196 | 484 | 289 | 121 | 0 |

The Concordance coefficient is calculated according to Equation (13) when there are no associated ranks:

$$W = \frac{12S}{n^2(m^3 - m)} = \frac{12 \cdot 2312}{8^2(9^3 - 9)} = 0.6020 \qquad (13)$$

Technological development is important for RTC freight transport, number $m > 7$. The Concordance coefficient is then calculated according to Equation (14) to obtain a random variable:

$$\chi^2 = n(m-1)W = \frac{12S}{nm(m+1)} = \frac{12 \cdot 2312}{8 \cdot 9(9+1)} = 38.533 \qquad (14)$$

The $\chi^2$ calculated value of 38.5333 is larger than the critical (equals 15.5073) value. As a result, the opinion of the responding experts is considered to be consistent, and the average ranks indicate the overall opinion of the experts [76].

According to Equation (15), the lowest value of the Concordance coefficient $W_{\min}$ is calculated:

$$W_{\min} = \frac{\chi^2_{v,\alpha}}{n(m-1)} = \frac{15.5073}{8(9-1)} = 0.242302 < 0.60208. \qquad (15)$$

If $W_{\min} = 0.242302 < 0.60208$, then the opinions of all eight respondents on the criteria for the evaluation of the technological development elements of the RTC are still considered reconciled. These criteria are important for the impact of RTC technological development tools on freight logistics cooperation.

Indicators of the importance of the impact of RTC technological development on freight transport for logistical cooperation are calculated: $Q_j$. The obtained data and all RTC technological development criteria and their order from the most important to the least important are presented in Table 7.

**Table 7.** Results of Ranking the Importance of RTC Technological Development Impact on Freight Transport Cooperation in Logistics (compiled by the Authors).

| Indicator Marker | Criterion Encryption Symbol ($m = 9$) | | | | | | | | | Sum |
|---|---|---|---|---|---|---|---|---|---|---|
| | **K1** | **K2** | **K3** | **K4** | **K5** | **K6** | **K7** | **K8** | **K9** | |
| $q_j$ | 0.1694 | 0.1472 | 0.1278 | 0.1778 | 0.0722 | 0.0500 | 0.0639 | 0.0806 | 0.1111 | 1 |
| $d_j$ | 0.8306 | 0.8528 | 0.8722 | 0.8222 | 0.9278 | 0.9500 | 0.9361 | 0.9194 | 0.8889 | 8 |
| $Q_j$ | 0.1038 | 0.1066 | 0.1090 | 0.1028 | 0.1160 | 0.1188 | 0.1170 | 0.1149 | 0.1111 | 1 |
| $Q_j'$ | 0.0528 | 0.0750 | 0.0944 | 0.0444 | 0.1500 | 0.1722 | 0.1583 | 0.1417 | 0.1111 | 1 |
| Distribution of importance of criteria | 8 | 7 | 6 | 9 | 3 | 1 | 2 | 4 | 5 | − |

Based on expert assessments and calculations, the criteria for RTC technological development elements to be assessed for freight transport in logistics cooperation are arranged, and five main criteria are presented:

1. Incompatibility of technology between road transport and other modes of transport;
2. Compatibility of collaborative technological developments of several RTCs;
3. Technological literacy of employees;
4. Ignorance of new technology management;
5. Lack of funds for technological development.

The analysis of the research results revealed that the elements of Road Transport Companies' technological development are related to assuring the efficient operation of transport logistics cooperation.

The calculated Kendall concordance coefficient does not identify those experts whose evaluations may differ from others. The competence coefficient [71] is calculated according to Formulas (7)–(10).

In this respect: $K_j^0 = \frac{1}{8} = 0.125$. The sums of the initial values in the columns of Table 6 are then multiplied by the initial competency coefficient. Group estimates of alternatives (Table 8) and a new matrix for calculating the competence factor were obtained. To calculate the final Kendall expert competence coefficients, the sum of each row of the matrix is divided by lambda (see Formula (9)), the size of which is 2089. It is important to note that the sum of the competence estimates thus calculated must be equal to one. According to the analysis and the obtained results in Table 8, the second, the fourth, the sixth and eighth expert have the highest (equal) competence compared with all the experts who participated in the survey.

**Table 8.** Competence coefficients of experts of importance of technological development elements of a road transport company (compiled by the authors).

| Expert Competence Coefficients | | | | | | | |
|---|---|---|---|---|---|---|---|
| E1 | E2 | E3 | E4 | E5 | E6 | E7 | E8 |
| 0.0998 | 0.1296 | 0.1239 | 0.1296 | 0.1292 | 0.1296 | 0.1284 | 0.1296 |

To check that opinions of all experts are competent, calculate according to the formula $\overline{K}_i^t - 1.96s \leq K_i^t \leq \overline{K}_i^t + 1.96s$ $\overline{K}_i^t$—the average of the competence coefficients; s—is the standard deviation and obtain intervals [0.0935; 0.1565]. The competence of the first expert in this group of experts is the lowest (0.0998), but not so low that the expert assessment should be eliminated during the research. In summary, it can be stated that the experts with the highest length of service in managerial positions for more than five years and the competence coefficients were the same 0.1296. Other experts had a sufficient level of competence to take their assessments into account.

According to Podvezko [61], direct and indirect evaluation methods' accuracy is higher than the ranking method's. The results of the evaluations are presented in Table 9.

The calculation of the significance of the criteria shows how the experts evaluate each of the criteria. Based on expert assessments and calculations, the significance of the criteria is presented in Table 10.

**Table 9.** Analysis of the determination of the significance of the criteria (compiled by the Authors).

| Expert | Significance of Criterion Evaluation | | | | | | | | |
|---|---|---|---|---|---|---|---|---|---|
| | K1 | K2 | K3 | K4 | K5 | K6 | K7 | K8 | K9 |
| E1 | 13 | 18 | 12 | 8 | 30 | 9 | 4 | 5 | 1 |
| E2 | 3 | 5 | 3 | 7 | 10 | 30 | 20 | 15 | 7 |
| E3 | 5 | 5 | 5 | 5 | 30 | 5 | 10 | 15 | 20 |
| E4 | 0 | 0 | 0 | 0 | 15 | 30 | 25 | 20 | 10 |
| E5 | 0 | 0 | 5 | 0 | 15 | 30 | 20 | 20 | 10 |
| E6 | 0 | 0 | 0 | 0 | 15 | 30 | 25 | 20 | 10 |
| E7 | 5 | 5 | 5 | 5 | 10 | 20 | 25 | 15 | 10 |
| E8 | 5 | 5 | 5 | 5 | 10 | 30 | 20 | 15 | 5 |
| sum: | 31 | 38 | 35 | 30 | 135 | 184 | 149 | 125 | 73 |
| Materiality of the criterion | 0.0386 | 0.0475 | 0.0438 | 0.0375 | 0.16875 | 0.23 | 0.1863 | 0.1563 | 0.0913 |

**Table 10.** Values of the significance of criteria (compiled by the authors).

| The Criterion | Materiality of the Criterion | | | | | | | | |
|---|---|---|---|---|---|---|---|---|---|
| | K1 | K2 | K3 | K4 | K5 | K6 | K7 | K8 | K9 |
| Place by significance | 8 | 6 | 7 | 9 | 3 | 1 | 2 | 4 | 5 |

Five criteria of RTC technological development were defined as the main (Figure 5):

K6—Incompatibility of technology connecting road transport and other modes of transport;
K7—The compatibility of technological development of cooperating Road Transport Companies;
K5—Technological literacy of employees;
K8—Ignorance of new technology management;
K9—Lack of funds for technological development.

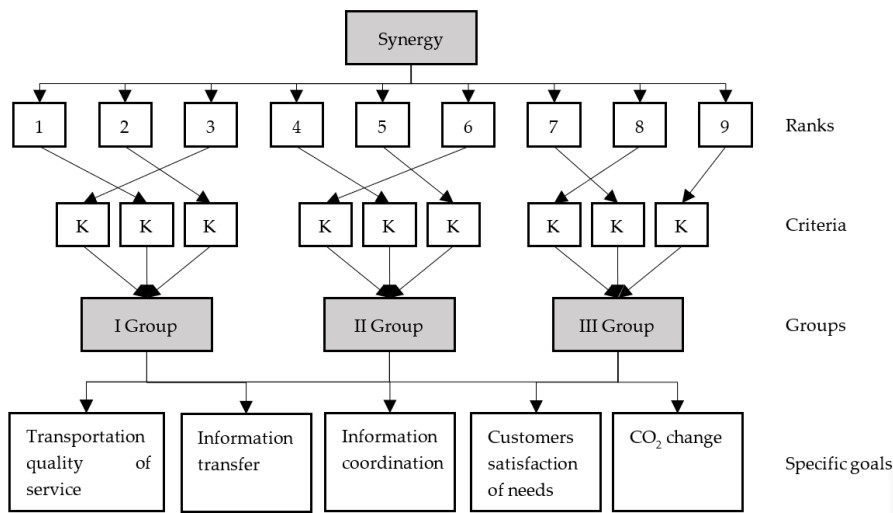

**Figure 5.** Algorithm of AHP performing (compiled by the authors).

The applied algorithm (methodological tree) for AHP performance is presented in Figure 5.

The realisations of the synergies of the cooperation of road transport companies depend on a technological development scope and compatibility of technology connecting (Figure 5).

The integration of road transport company technological development elements that are defined as very important for freight transport in logistical cooperation is an essential key condition for the emergence and growth of synergy between technological development and logistical cooperation. New identified four factors of synergy between technological development and logistical cooperation are shown in the blue colour background in Figure 6.

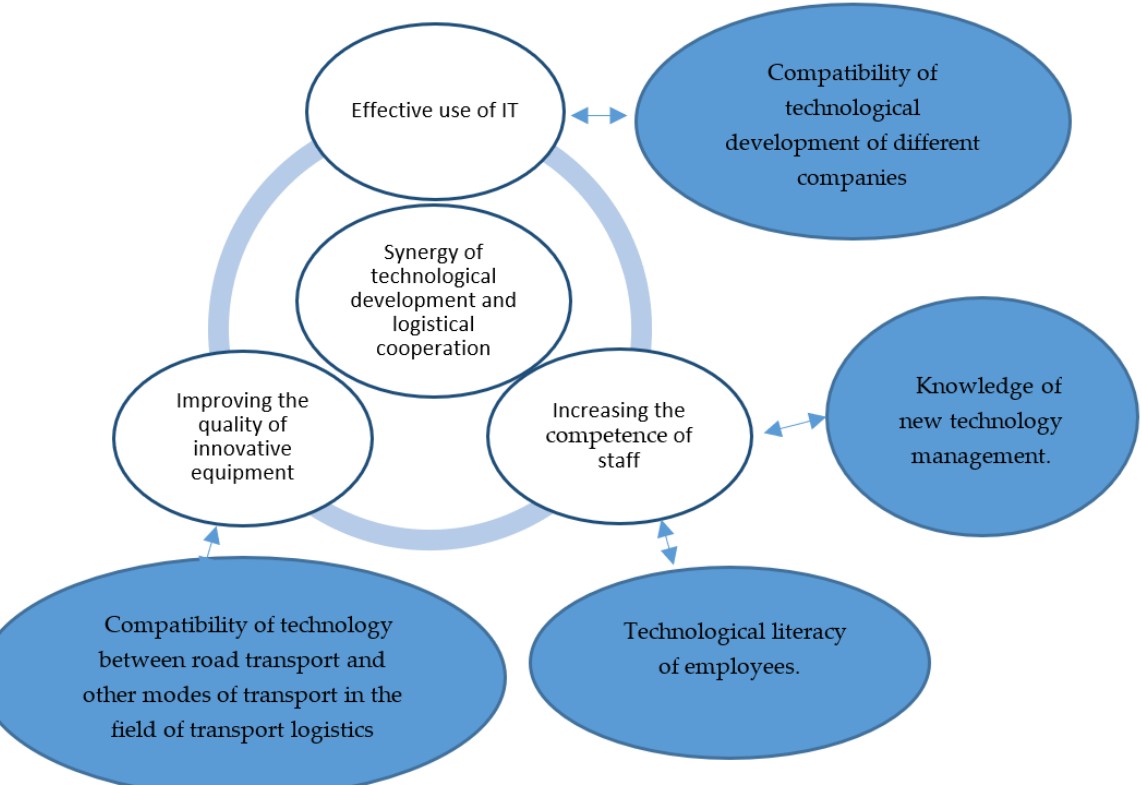

**Figure 6.** New defined factors of synergy between technological development and logistical cooperation (compiled by the authors).

The advantage of factors determined by the authors is that by not destroying the before-identified interaction system, they perceptibly supplement the notion of RTC technological development and simultaneous logistic cooperation synergy phenomenon. These four factors are (Figure 6):

1. Compatibility of technology between road transport and other transport modes in transport logistics cooperation;
2. Compatibility of technological development of different companies;
3. Technological literacy of employees;
4. Knowledge of new technology management.

In summary, the most significant criterion is the incompatibility of the IT technologies of road and non-road transport systems. The compatibility of the technological development of the cooperating road transport undertaking is quite important as well (Figure 6). According to the experts, employees' limited technological literacy and ignorance of new technology management are also fairly significant.

## 5. Discussion

The achievements of road transport companies in the supply chain depend on a well-managed strategy for cooperation and technological development, a process in which technological development plays a rather important role. Hong and Nguyen [77] argued that the significant role of collaboration between RTCs and technological capabilities in achieving effective collaborative strategies has always been viewed in parallel. Technology development during a period of cooperation between road transport companies is usually studied from a socio-economic perspective. The company's technological development is associated with updating and improving technologies. Improvements are complex processes related to major organisational systems, processes, products or industry challenges. Road transport companies ensure the development of new areas of activity or services through cooperation in developing technologies and implementing continuous renewal processes. By cooperating on technologies, sharing knowledge and developing new competencies of employees, companies ensure appropriate technological possibilities for the development of activities. By mutually developing the necessary customer attraction mechanisms and promoting supplier satisfaction, companies receive preferential access to suppliers' information resources. Noticeably, there is a lack of research work on evaluations of the issue of employee action and computer engineering literacy. During the period of the COVID-19 pandemic, the importance of the synergy between technological development and logistical cooperation of road transport companies has especially increased. 2020–2021 The period of COVID-19 pandemic has highlighted that the transportation of production goods and the use of existing information systems in collaboration is problematic. Ensuring the coherence of technological development and the need for innovative technologies in the logistics cooperation of companies has increased. When determining road transport companies' investment in technological development during cooperation, process interaction criteria directly impact the intensity of road transport cooperation synergies. The research carried out by the authors of this article has revealed the impact of technology development on the synergy between logistics cooperation and road transport company activities. The obtained research results are applicable as a basis for the cooperation agreement between logistics and road transport companies, i.e., RTC decision-makers have gained the tools to regulate and prepare the sphere of technological development possibilities and applications. It should be emphasised that by applying technological development during cooperation, RTCs reduce customer service expenses and, at the same time, speed up the processes carried out during cooperation.

The question of the reliability and uncertainty of the obtained results arises because, during these studies, transport and logistics companies of different sizes were not interviewed. The interviewed experts represented only the big Lithuanian carriers. The level of IT provision, the applied management system and the logistics volume significantly depend on the size of the companies. Thus, the collected opinions of experts are quite subjective.

Another factor limiting the completeness of the research is that the environmental impact of environmentally friendly engines (electric, hydrogen-powered cells, solar-charged cells) and drones was not analysed when evaluating the technological development of the transport sector. Implementing these innovative technologies requires significant additional financial resources from companies and therefore is an essential factor in decision-making.

## 6. Conclusions

Using the AHP algorithm, the evaluation methodology to assess the synergy between the technological development of a road transport company and logistics lines is developed and realised.

According to interviewed experts, continuous technological development is indispensable to ensure transport companies' efficient operation and sufficient competitiveness. The defined four decisive factors (compatibility of technology between road transport and other modes of transport in the field of transport logistics cooperation, compatibility of technological development of different companies, technological literacy of employees and

the knowledge of new technology management) mostly impact increasing the synergy between the technological development and logistical cooperation considering the efficiency of the use of information technologies. By not destroying the before-identified perceptible interaction system, these factors enhance the notion of RTC technological development and logistic cooperation synergy phenomenon.

The analysis of the synergetic factors of technological development and logistics cooperation revealed that the compatibility of technological systems of road transport and other modes of transport and the technological development of different companies ensures the breakthrough of a larger flow of information. The lack of funds hinders technological development for freight transport management as the continuous high-tech progress ensures the timely introduction of new technologies in companies.

The compatibility of the technological development of road transport companies and the synergy of logistics cooperation is an indispensable factor in managing the increasing information flows and generated operational data in the logistical cooperation. The interlink of the updated technology of road transport companies with the transfer of information processed by logistical cooperation is neither timely nor sufficient. The transfer of up-to-date information on incompatibilities of technologies between transport companies is unavoidable. So, the outcome of this research confirmed the synergy of technological development of transport companies in the course of logistical cooperation. Therefore, it is necessary to allocate funds for at least a minimal provision of IT tools for the synchronisation of technologies between cooperating companies. The technological development of transport companies is strongly dependent on the professional development of company employees. Consequently, it is compulsory to maintain the appropriate level of technological literacy of relevant employees.

The authors intend to deepen their oncoming further research by applying the multicriteria evaluation methods. Furthermore, it is planned to use artificial intelligence tools for simulating non-standard (e.g., pandemics) and unstable geopolitical situations that perceptibly restrict mobility. Analogous research on the synergy between technological development and logistics is expected to be carried out in other modes of transport: railway, sea and air transport sectors.

**Author Contributions:** Conceptualisation, K.V., A.K. and G.B.; methodology, K.V. and G.B.; software, K.V. and A.K.; validation, K.V., A.K. and G.B.; formal analysis, A.K., G.B. and K.V.; investigation, A.K. and G.B.; resources, K.V.; data curation, K.V.; writing—original draft preparation, K.V. and A.K.; writing—review and editing, G.B.; visualisation, K.V.; supervision, G.B.; project management and final revising of the text; K.V. All authors have read and agreed to the published version of the manuscript.

**Funding:** This research received no external funding.

**Institutional Review Board Statement:** The study was conducted in accordance with the Declaration of Helsinki, and approved by the Institutional Review Board (Ethics Committee) of Vilnius Gediminas Technical University (protocol code 10.6-07-10.21-11519; 25 November 2021).

**Informed Consent Statement:** Informed consent was obtained from all subjects involved in the study.

**Data Availability Statement:** Not applicable.

**Conflicts of Interest:** The authors declare no conflict of interest.

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
