# Peer review of "The Synergy between Technological Development and Logistic Cooperation of Road Transport Companies"

_sustainability, doi:10.3390/su142114561_

Round 1

Reviewer 1 Report (Previous Reviewer 2)

The manuscript is about Synergy between Technological Development and Logistic Cooperation of Road Transport Companies. This is quite an important topic, especially in the period of major turbulences in world logistics caused by the Covid-19 pandemic. However, the text of the article does not refer to the significant changes that took place in the logistics sector during the pandemic. Although this was not the purpose of the article, I consider the complete omission of this issue a significant drawback. This proves that the most recent events, which had a significant impact on the analyzed sector of the economy, were not taken into account. In order to correct this drawback, even a short fragment concerning changes in the logistics sector in recent years should be completed.

Hypothesis 3 should be clarified. As it stands, it contains a questionable formulation (maybe) that is difficult to verify unequivocally. It also influences the possibility of professional confirmation or falsification.

The structure of the article is disproportionate.

Chapter 6. Discussion only contains 2 paragraphs. It is definitely too little to conduct a professional scientific discussion of the results of the analyzes. This must be completed.

In diagram 4 (page 11), it is unnecessary to include its title in the diagram field, if it is written in the title (repeating information).

Author Response

Dear Reviewer,

We appreciate your precious time in reviewing our paper and providing valuable comments. The authors have carefully considered the comments and tried our best to address every one of them. We hope the manuscript after careful revisions meet your high standards.  All modifications in the manuscript have been highlighted.

Reviewer 2 Report (Previous Reviewer 1)

While there are not much significant changes that improve the readability and quality of the manuscript, I recommend a proofreading of a native speaker with a language check before publication.

Author Response

Dear Reviewer,

We appreciate your precious time in reviewing our paper and providing valuable comments. The authors have carefully considered the comments and tried our best to address every one of them. We hope the manuscript after careful revisions meet your high standards.  All modifications in the manuscript have been highlighted.

Reviewer 3 Report (New Reviewer)

The paper entitled Synergy between Technological Development and Logistic Cooperation of Road Transport Companies deals with the actual topic. However, this paper has major lacks:

·        The abstract is not well written. The most important results/findings must be listed.

·       Keywords are not well defined (for example transport company management; freight transport logistics chain; sustainable supply chain innovation). They must be shorter, clearer, and more precise.

·       Section 3 title is Technological development of the transport company and efficiency of freight transport logistics. The efficiency of freight transport is very complex, and different methods are used for measuring and improving it. In this chapter, you don’t deal with efficiency?!

·       By the way you have two Sections 3!?

·       Also, the term “freight transport logistics” is contested.

·       The argumentation for the AHP method is not good. (lines 242 -245). There are different methods with the same and additional advantages.

·       Proposed model is trivial (figure 6).

·       In that manner, all results are questionable and relative.

  • The separate section Practical and theoretical implications (or Discussion) is missing. The existing section Discussion is very modest. This confirms the lack of scientific and practical contributions.
  • The conclusion section is not on a satisfactory level. Clearly state your unique research contributions in the conclusion section. Future research directions are missing.
  • Limitations of this research are also missing.

·       Scientific contributions are questionable (bearing aforementioned in the mind there is no scientific justification): “The purpose of this research work is to justify scientifically the influence the synergy on the efficiency of technological development and logistic cooperation of RTC”.

·       In a technical manner authors uploaded manuscript with text which is marked with different colors. I am not sure if your last version for review?! (line 20, line 61, line 84, line 223, line 342, etc). 

Author Response

Dear Reviewer,

We appreciate your precious time in reviewing our paper and providing valuable comments. The authors have carefully considered the comments and tried our best to address every one of them. We hope the manuscript after careful revisions meet your high standards.  All modifications in the manuscript have been highlighted.

Reviewer 4 Report (New Reviewer)

Review Comments on

“Synergy between Technological Development and Logistic Cooperation of Road Transport Companies”

In this paper, the technological development of road transport companies is analysed and evaluated according to its impact on the company’s performance. My comments on the paper are presented below:

Abstract

The abstract section can be written well. It would be better to rewrite this section. Establishment of the research topic, main objective, and new findings should be presented clearly. Sentence formation is also incorrect. Two sentences are started with “During”. Please rewrite those sentences also. Also include the major findings. In addition to that, also check the key words and modify the key words. Consider only the relevant key words.

Introduction

This section lacks the motivation of research. Some industry examples should be added in the support of the research questions mentioned. Discussion should be in a flow rather than individual discussion.

Please check the full manuscript. Refine the language and eliminate the spelling errors. Basic English sentence formation is also incorrect in some places. (Like, the first sentence of the first paragraph of Introduction Sentence).

Add Literature Review Section. This section should be written in two or three sub-sections. First sub-section should consider the research papers based on Technological Development and Logistic. Second sub-section should be based on Methodology.

Add following research papers to update and modify literature review section

i. Analytic hierarchy process: An overview of applications

ii. Decision-making framework for identifying regions vulnerable to transmission of COVID-19 pandemic.

iii. A fuzzy based hybrid decision-making framework to examine the safety risk factors of healthcare workers during COVID-19 outbreak.

iv. Analysis of barriers to implement drone logistics.

Analysis of technological development processes of road transport companies

• Update the title of the figure 1.

• Include the section 3 in the literature review. As at the end of this section, it is mentioned that the based on the literature review……

3. Methodology for assessing the impact of technological development on logistical cooperation

• This section should be renamed with methodology section. Explain AHP methodology also in this section.

4. Results of determination of technological development impact on logistical cooperation

• This section should be renamed as Results.

5. Evaluation methodology

• This section should come before the results section. Please modify and update.

Author Response

Dear Reviewer,

We appreciate your precious time in reviewing our paper and providing valuable comments. The authors have carefully considered the comments and tried our best to address every one of them. We hope the manuscript after careful revisions meet your high standards.  All modifications in the manuscript have been highlighted.

Round 2

Reviewer 3 Report (New Reviewer)

Unfortunately, I believe that this paper is not for publication.

As I said in the first round, the main objections still stand.

The paper does not have the necessary scientific contributions.

Reviewer 4 Report (New Reviewer)

All the comments are addressed properly. 

This manuscript is a resubmission of an earlier submission. The following is a list of the peer review reports and author responses from that submission.

Round 1

Reviewer 1 Report

Thanks for your nice efforts on preparation of this study. There is a good effort to propose a method to analyze the technological development and logistic co-operation of road transport companies in this study. However there are some points to needs improvement before publication.

- The paper needs an introduction part. The paper starts with a literature review in the first part, this is not seem good to understand the motivation and research questions of the paper.

- Since Figure 1 and Figure 2 are compiled by authors, they can be presented in the proposed method or application parts of the study.

- The evaluation way presented by Table 1 is not a familiar thing with the AHP method mentioned in the previous paragraph. AHP consists pairwise comparison of decision elements i.e. comparisons among criteria, comparisons among alternatives.

- 1-9 scale presented in Table 2 is not compiled by authors, it is the scale of Saaty. Please provide appropriate citations.

- What are the column headings in Table 3? It must be given.

- It would be better if a flowchart for the application of the method.

I feel these revisions will be helpful for a better understanding of the study. 

Author Response

Thank You very much for your time and coments. All your comments are considered and corrected

Thank you very much for your time and comments. They are all corrected and marked

- The paper needs an introduction part. The paper starts with a literature review in the first part, this is not seem good to understand the motivation and research questions of the paper.

Thanks for the note. Considered and corrected.

- Since Figure 1 and Figure 2 are compiled by authors, they can be presented in the proposed method or application parts of the study.

Thanks for the note. Considered and corrected.

- The evaluation way presented by Table 1 is not a familiar thing with the AHP method mentioned in the previous paragraph. AHP consists pairwise comparison of decision elements i.e. comparisons among criteria, comparisons among alternatives.

Thanks for the note. Considered and corrected.

- 1-9 scale presented in Table 2 is not compiled by authors, it is the scale of Saaty. Please provide appropriate citations.

Thanks for the note. Considered and corrected.

- What are the column headings in Table 3? It must be given.

Thanks for the note. Considered and corrected.

- It would be better if a flowchart for the application of the method.

I feel these revisions will be helpful for a better understanding of the study. 

Thanks for the note. Considered and corrected.

Reviewer 2 Report

The article deals with quite an important problem Synergy between Technological Development and Logistic Cooperation of Road Transport Companies.

Due to the rising costs of transport and interrupted supply chains as a result of the pandemic, I consider the topic topical and important. High transport costs increase the prices of the offered products. The search for ways to reduce logistics costs can be one of the ways to limit the pace of price increases. Many countries in the world are currently struggling with the problem of high inflation.

The purpose of the article is missing in the abstract.

See page 13 for missing Figure 4. Model algorithm. This should be completed.

The structure of the manuscript is slightly disturbed. Chapter 3 combines the methodology with the description of the results obtained. In my opinion, these two elements should be separated and placed in separate chapters.

Due to the fact that the research was based on the opinions of only 8 experts, the method of their selection should be clearly indicated. Their number is low, so it must be clearly proved that their selection was correct and that the results can be representative.

The manuscript does not indicate the limitations of the methods used and the results obtained.

I recommend that you rethink hypothesis 3. It seems to be quite obvious as it stands. One can consider reformulating it.

Rather positively in the quantitative sense, I assess the use of the literature on the subject. In my opinion, however, there is no broader discussion of the results of my own research with other studies known from the literature. The use of the literature on the subject should be increased in the second part of the study.

The method of reference in the text to the cited items should be different. Not [16, 17, 18], but [16-18] (page 2).

What do the letters a-i in the Figure 3 (page 9) mean? This needs to be clarified.

Sometimes in the text (eg. pages 1, 2) there appear not necessary terms in scientific articles next to the names of the authors "Researcher". You can simply use the names of the authors cited.

Author Response

Thank you very much for Your time and comments. 
All Your comments are considered and corrected and marked.

Thank you very much for your time and comments. They are all corrected and marked

Due to the rising costs of transport and interrupted supply chains as a result of the pandemic, I consider the topic topical and important. High transport costs increase the prices of the offered products. The search for ways to reduce logistics costs can be one of the ways to limit the pace of price increases. Many countries in the world are currently struggling with the problem of high inflation.

The purpose of the article is missing in the abstract.

Thanks for the note. Considered and corrected.

See page 13 for missing Figure 4. Model algorithm. This should be completed.

Thanks for the note. Considered and corrected.

The structure of the manuscript is slightly disturbed. Chapter 3 combines the methodology with the description of the results obtained. In my opinion, these two elements should be separated and placed in separate chapters.

Thanks for the note. Considered and corrected.

Due to the fact that the research was based on the opinions of only 8 experts, the method of their selection should be clearly indicated. Their number is low, so it must be clearly proved that their selection was correct and that the results can be representative.

Thanks for the note. Considered and corrected.

The manuscript does not indicate the limitations of the methods used and the results obtained.

Thanks for the note. Considered and corrected.

I recommend that you rethink hypothesis 3. It seems to be quite obvious as it stands. One can consider reformulating it.

Thanks for the note. Considered and corrected.

Rather positively in the quantitative sense, I assess the use of the literature on the subject. In my opinion, however, there is no broader discussion of the results of my own research with other studies known from the literature. The use of the literature on the subject should be increased in the second part of the study.

Thanks for the note. Considered and corrected.

The method of reference in the text to the cited items should be different. Not [16, 17, 18], but [16-18] (page 2).

 Thanks for the note. Considered and corrected.

What do the letters a-i in the Figure 3 (page 9) mean? This needs to be clarified.

 Thanks for the note. Considered and corrected.

Sometimes in the text (eg. pages 1, 2) there appear not necessary terms in scientific articles next to the names of the authors "Researcher". You can simply use the names of the authors cited.

 Thanks for the note. Considered and corrected.

Reviewer 3 Report

Road transport companies are analyzed and evaluated, mainly, on the development of freight transport logistics services and competitiveness. The key elements of the freight transport logistics chain are identified. An Analytical Hierarchy Process (AHP) method is used to determine the influence of technological development on the efficient use of IT and equipment renewal. Questions about the influence of road transport technology development elements on the quality of freight transport logistics chain, according to 3 groups of criteria, are formulated and an expert evaluation questionnaire is created. Having processed the survey data of 8 experts, the research results are systematized and the criteria are arranged. Thus, technological development priorities are determined.

Certainly the applications is very interesting. However the method's presentation is quite defficient. In eq's (1)-(6), (8)-(10)  there are many undefined terms, e.g. S, \chi_{\nu,\alpha}, x_{ij}, K^t. The authors shoud recall the notation introduced in reference [56] of the reviewed paper. No hint is given on the calculation procedures for Tables 2-5.

The current version fails to introduce formally the calculation procedures, and it lists a table series without a clear description of the computing procedures.Hence the drawn conclusions of the paper cannot be taken as fully justified.

Author Response

Thank you very much for your time and comments. 
All your comments are considered and corrected and marked.

Thank you very much for your time and comments. They are all corrected and marked

Certainly the applications is very interesting. However the method's presentation is quite defficient. In eq's (1)-(6), (8)-(10)  there are many undefined terms, e.g. S, \chi_{\nu,\alpha}, x_{ij}, K^t. The authors shoud recall the notation introduced in reference [56] of the reviewed paper. No hint is given on the calculation procedures for Tables 2-5.

Thanks for the note. Considered and corrected.

The current version fails to introduce formally the calculation procedures, and it lists a table series without a clear description of the computing procedures.Hence the drawn conclusions of the paper cannot be taken as fully justified.

Thanks for the note. Considered and corrected.

Round 2

Reviewer 1 Report

The paper still needs an introduction part which tells the motivation of the research clearly, a clear explanation of the methodology and a better presentation of results. That is why I made my decision.

Reviewer 3 Report

The authors took care of the remarks in my former review. In the current version the used mathematical model is clearly exposed and the correspondence of the experimental results with the model is now appreciated. The main contribution of the paper is certainly the application of the exposing model within real transport analysis.